# Mini-Omni: Language Models Can Hear, Talk While Thinking in Streaming

**Zhifei Xie**♠♣*
xzf24@mails.tsinghua.edu.cn

## Abstract

Recent advances in language models have achieved significant progress. GPT-4o, as a new milestone, has enabled real-time conversations with humans, demonstrating near-human natural fluency. Such human-computer interaction necessitates models with the capability to perform reasoning directly with the audio modality and generate output in streaming. However, this remains beyond the reach of current academic models, as they typically depend on extra TTS systems for speech synthesis, resulting in undesirable latency. This paper introduces the **Mini-Omni**, an audio-based end-to-end conversational model, capable of real-time speech interaction. To achieve this capability, we propose a text-instructed speech generation method, along with batch-parallel strategies during inference to further boost the performance. Our method also helps to retain the original model's language capabilities with minimal degradation, enabling other works to establish real-time interaction capabilities. We call this training method **"Any Model Can Talk"**. We also introduce the **VoiceAssistant-400K** dataset to fine-tune models optimized for speech output. To our best knowledge, **Mini-Omni** is the first fully end-to-end, open-source model for real-time speech interaction, offering valuable potential for future research.

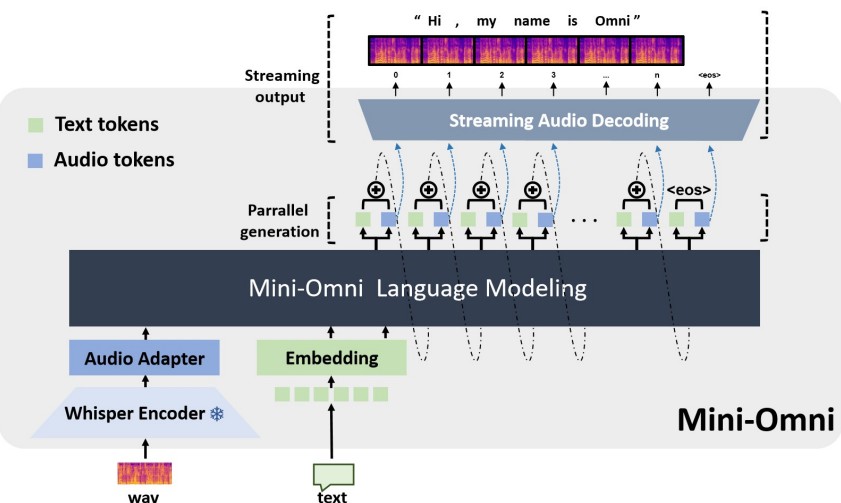

Figure 1: The **Mini-Omni** model architecture.

---

*equal contribution

Technical report.

# 1 Introduction

Recent developments in large language models have progressed rapidly, with models becoming increasingly powerful, such as off-the-shelf Llama 3.1 [meta, 2024], Mixtral [mixtral, 2024], Qwen-2 [Yang et al., 2024], and the well-known GPT-4. As an extension of their capabilities, language models are beginning to master understanding other modalities, exemplified by LLaVA [Liu et al., 2024], Qwen2-Audio [Chu et al., 2024] and Video-llama [Zhang et al., 2023b]. Despite their strength in specific tasks, a significant gap remains that hinders further integration of large language models into daily application: real-time voice interaction capability. GPT-4o [openai, 2024], introduced by OpenAI, is the first model to feature real-time multimodal speech interaction capabilities. It can understand and engage with vision, audio, and text while enabling real-time speech conversations, although it remains closed-source. Other models typically adopt two approaches to incorporate speech capabilities. The first is a cascade method, where the language model generates text, followed by a text-to-speech (TTS) model for audio synthesis. This approach introduces significant latency due to the time required for text generation, severely impacting user experience. The second, an end-to-end method like SpeechGPT [Zhang et al., 2023a], generates text before continuing to generate audio. However, this still requires waiting for text generation. Large language models need real end-to-end speech output capabilities to provide real-time feedback.

Enhancing models with speech output capabilities is a challenging task, primarily due to four factors: (1) **Complexity of Audio Reasoning**: Our experiments indicate that direct training for audio modality reasoning is highly challenging, often resulting in incoherent outputs from the model. (2) **Model Complexity**: Incorporating additional modules for speech input and output increases the overall complexity. (3) **Difficulty in Modality Alignment**: The reasoning abilities developed for text are difficult to transfer to the audio domain. (4) **Resource Demands**: Adapting a model's text capabilities to the speech modality requires converting all data labels into audio and retraining, significantly increasing resource consumption.

In this paper, we propose **Mini-Omni**, the first open-source multi-model large language model with real-time conversational capabilities, featuring fully end-to-end speech input and output abilities. It also includes various other audio-to-text functionalities such as Automatic Speech Recognition (ASR). We adapt currently available off-the-shelf methods for discretizing speech tokens and employ the simplest model architecture, making it easy for our model and approach to be adapted by other researchers. Direct audio reasoning poses significant challenges; however, our approach successfully addresses this using only a 0.5B model and a limited amount of synthesized audio data. Importantly, our training framework achieves this without heavy reliance on extensive model capabilities or large volumes of data.

To leverage and preserve the original capabilities of the language model, we propose a parallel generation paradigm in which the transformer simultaneously produces audio and text tokens. Subsequently, we observed a minimal impact of the audio modality on text capabilities and further introduced **batch-based parallel generation**, which significantly enhances the model's reasoning ability during streaming audio output. As a poinerr, we opted not to sacrifice audio quality for a simpler and lower bitrate audio encoder, in order to reduce the complexity of audio inference in the model. However, to ensure audio quality, we selected SNAC [Siuzdak, 2024], a music-grade encoder features 8 layers of codebooks and processes hundreds of tokens per second. Innovatively, we applied **text-instructed delayed parallel generation** to address the issue of long SNAC codebook sequences. Experiments show that the audio output quality is on par with common TTS systems.

We also propose a method that requires minimal training and modification of the original model, enabling other works to rapidly develop their own speech capabilities. We refer to this approach as **"Any Model Can Talk"**, designed to achieve speech output using a limited amount of additional data. The approach extend speech capabilities through additional adapters and pre-trained models, fine-tuning with a small amount of synthesized data. This is combined with the aforementioned parallel modeling approach to enable streaming output in the new modality while preserving the original model's reasoning capabilities.

To evaluate the capabilities of **Mini-Omni**, we first assessed its performance on traditional text-to-speech multi-modal tasks, including text-based question answering (textQA), automatic speech recognition (ASR), text-to-speech response, and speech-based question answering (speechQA). The model demonstrated strong proficiency in these fundamental tasks. Additionally, we conduct a series of experiments to investigate the impact on the original model's capabilities and assess the

effectiveness and variations of our inference method. Preliminary experiments demonstrate that batch parallel inference preserves the model's original capabilities. We will conduct further experiments and provide additional details in due course.

Lastly, we observed that most open-source QA datasets contain mixed code or overly lengthy text, rendering them unsuitable for speech model. To overcome this limitation, we introduce the **VoiceAssistant-400K** dataset, comprising over 400,000 entries specifically generated by GPT-4o for speech assistant supervised fine-tuning (SFT).

**In summary, we make the following contributions:**

- We introduce **Mini-Omni**, the first open-source end-to-end multimodal large model with audio input and audio streaming output capabilities. We propose a unique text-instruct parallel generation method that enables speech inference outputs aligned with textual capabilities, achieved with minimal data. We further enhance this with delayed parallelism, accelerating audio inference speed.

- We introduce "**Any Model Can Talk**", an innovative approach that enhances performance without altering the architecture of large models by focusing on training and inference. Our method employs a three-phase training process for speech-to-text and text-to-speech adapters, including annealing and SFT. Our method involves minimal training and modification of the original model, aiming to provide a reference for incorporating interaction capabilities into other models.

- We identified shortcomings in existing open-source QA datasets when training audio assistants and proposed a dedicated dataset for speech model outputs, called **VoiceAssistant-400K**. This dataset, synthesized using GPT-4o, can be used to fine-tune models to develop the tone of a voice assistant.

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
