# OpenReview forum: "Mini-Omni: Language Models Can Hear, Talk While Thinking in Streaming"
_tsinghua.edu.cn/THU/2024/Fall/AML — THU 2024 Fall AML Submission_

### Official Review · ~Lily_Sheng1 · 2024-11-08
**Submission 26 Review**

**Rating:** 9
**Confidence:** 4

**Review:**

The Mini-Omni project presents a novel approach to enabling end-to-end real-time conversational capabilities in a large language model, allowing multimodal interaction by incorporating both audio input and streaming audio output.

Pros:
1. The model's design focuses on a lightweight 0.5B architecture and a limited amount of synthesized audio data, allowing robust multimodal capabilities without heavy computational demands.
2. The VoiceAssistant-400K dataset fills in a gap in current open-source resources.

Cons:
1. The VoiceAssistant-400K dataset uses GPT-4o for generation which may introduce bias or repetitive phrasing patterns.
2. There could be more details on the methodology and approaches.

---

### Official Review · ~Zhen_Leng_Thai1 · 2024-11-08
**Promising Proposal for Real-Time Audio-Based Conversational Model**

**Rating:** 9
**Confidence:** 4

**Review:**

This paper proposes an audio-based end-to-end conversational model for real-time interaction, with a solid training method and dataset. However, the introduction contains a typo ("poinerr"), the structure requires improvement, and the task definition could be more clearly articulated.

---

### Official Review · ~Joydeep_Chandra2 · 2024-11-08
**The title "Mini-Omni: Language Models Can Hear, Talk While Thinking in Streaming" is clear and accurately reflects the core idea of the project, highlighting its focus on real-time, multimodal capabilities.**

**Rating:** 8
**Confidence:** 4

**Review:**

The proposal is strong, with a clear problem definition, with an innovative methodology, and high potential impact. It demonstrates a thorough understanding of the field and offers a novel solution to a relevant problem. However, it could provide more specific details on implementation plan and specifying evaluation metrics. The grammatical structure also could be reviewed.

---

### Official Review · ~Aleksandr_Algazinov1 · 2024-11-09
**Innovative and relevant**

**Rating:** 10
**Confidence:** 4

**Review:**

The proposal is well-written and clearly defines the problem. Authors propose a new (Mini-Omni) model for real-time speech interaction. The motivation is clearly explained and leaves no doubt about the relevance of the study. Besides the new model, a new training approach (Any Model Can Talk) and a new dataset (VoiceAssistant-400K) are introduced in the proposal. This is impressive and indicates the high potential of the project.

---

### Official Review · ~Yufei_Zhuang1 · 2024-11-09
**Exciting and highly relevant**

**Rating:** 9
**Confidence:** 4

**Review:**

This paper presents an exciting and highly relevant research in the field of language models. The recognition of the limitations in current academic models, especially the latency issue caused by relying on additional TTS systems for speech synthesis, is astute.
The introduction of Mini - Omni as an audio - based end - to - end conversational model capable of real - time speech interaction is a significant step forward. The proposed text - instructed speech generation method, along with the batch - parallel inference strategies, shows innovation. These not only enable real - time interaction but also manage to maintain the original language capabilities of the model with minimal degradation, which is a remarkable achievement.
The concept of "Any Model Can Talk" training method has great potential as it allows other works to build real - time interaction capabilities more easily. Additionally, the introduction of the VoiceAssistant - 400K dataset for fine - tuning models optimized for speech output further enriches the research. Overall, Mini - Omni being the first fully end - to - end, open - source model for real - time speech interaction opens up valuable opportunities for future research in human - computer conversation and related fields.

---

### Official Review · ~Yuanda_Zhang1 · 2024-11-09
**interesting idea**

**Rating:** 9
**Confidence:** 4

**Review:**

This work proposes Mini-Omni, which is the first fully end-to-end, open-source model for real-time speech interaction. Meanwhile, it offers potential for other models to incorporate interaction capabilities. Given the information from the proposal, we have that Mini-Omni will be a efficient multmodal large model with high speed of inference, that's quite interesting and promising in real market. However, more details about approaches and implementation need to be provided.

---

### Official Review · ~Diego_Cerretti1 · 2024-11-10
**Interesting and relevant**

**Rating:** 9
**Confidence:** 4

**Review:**

The authors propose an open-source, end-to-end system for real-time audio-based interaction. This proposal addresses the necessity for models that can smoothly handle real-time speech. Overall, the proposal is an interesting contribution to multimodal model research, exhibiting great practical relevance.

---

### Official Review · ~Rim_El_Filali1 · 2024-11-11
**Promising Open-Source Real-Time Speech Model with Evaluation Gaps**

**Rating:** 8
**Confidence:** 4

**Review:**

This paper presents Mini-Omni, an innovative open-source, multimodal language model with real-time speech interaction capabilities, bridging a gap in current language models’ abilities to engage in real-time audio conversations. Additionally, it introduces a specialized dataset, VoiceAssistant-400K, specifically tailored for training speech assistants.

Pros:
- Batch-parallel inference and delayed parallel generation effectively reduce latency, enhancing real-time interaction.
- The "Any Model Can Talk" training method adds audio capabilities to models with minimal overhead.

Cons:
- The proposal lacks a clear flow in places, particularly in separating high-level objectives from technical details, which may hinder readability.
- More empirical data on performance benchmarks would enhance the paper’s impact.

---

### Official Review · ~Ziyu_Zhao6 · 2024-11-11
**Review of "Mini-Omni: Language Models Can Hear, Talk While Thinking in Streaming" Proposal**

**Rating:** 9
**Confidence:** 4

**Review:**

Overview:
This proposal introduces Mini-Omni, an open-source large language model with end-to-end real-time speech interaction capabilities. Mini-Omni is designed to handle both speech input and output through a parallel generation process, enabling audio responses alongside textual outputs. Additionally, the proposal includes the VoiceAssistant-400K dataset, specifically created for training voice assistants, which addresses limitations in current open-source datasets for audio models.

Strengths:
	1.Innovative Contribution to Multimodal LLMs: Mini-Omni stands out as the first open-source, fully end-to-end model capable of real-time audio interaction. This innovation fills a significant gap in multimodal LLMs and offers a new direction for open-source research in audio-based AI interactions.
	2.Low Resource Requirement for Adaptation: The “Any Model Can Talk” approach allows other models to integrate speech capabilities with minimal additional data and training, making it accessible to a broader range of researchers and developers.
	3.Dedicated Speech Dataset: The VoiceAssistant-400K dataset addresses the shortcomings in existing QA datasets for speech-based tasks. This specialized dataset is valuable for training high-quality voice assistants and can contribute broadly to research on audio-enabled LLMs.

Weaknesses:
Reliance on Synthesized Data from GPT-4o: The reliance on VoiceAssistant-400K for fine-tuning could lead to limitations in handling nuanced or naturalistic speech patterns, potentially impacting the model’s adaptability in real-world scenarios.

---

### Official Review · ~Michael_Hua_Wang1 · 2024-11-11

**Rating:** 8
**Confidence:** 3

**Review:**

The proposal describes Mini-Omni, an open-source model permitting humans to have real-time voice conversations with LLMs with minimal latency.

On the face of it, this project is quite ambitious, and if successful, it certainly stands to contribute to research in the area. However, the scope of the work to actually be done is unclear, given that the author appears to have already submitted a preprint of this exact research topic to Arxiv a few months ago. Is the proposal here to do additional refinement following the methodology previously used for the pre-print, or is there additional work to be done?

This proposal could stand to clearly delineate the scope of the work with the above in mind.

---

### Official Review · ~jin_wang30 · 2024-11-12
**a surprising and promising proposal**

**Rating:** 10
**Confidence:** 4

**Review:**

This proposal introduces Mini-Omni, an open source end-to-end multimodal language model that supports real-time voice interaction. The model aims to achieve real-time human-computer voice conversation through two-way interaction of audio and text, solve the delay problem of current mainstream language models in voice processing, and proposes an innovative "Any Model Can Talk" training method.

Advantages：
Mini-Omni is the first open source multimodal language model that supports real-time voice input and output. It realizes voice generation in parallel with text output through the "Any Model Can Talk" method, providing a groundbreaking reference for future multimodal human-computer interaction research. In addition, the model solves the delay problem of traditional text generation and then conversion to voice through a parallel reasoning strategy, making the voice interaction experience smoother. It also introduces the VoiceAssistant-400K dataset dedicated to voice assistants, and optimizes the model using techniques such as delayed parallel generation and batch reasoning to ensure that the efficiency of voice generation is improved while maintaining text capabilities.
I believe that the architecture of Mini-Omni has the potential to adapt to other research, allowing for rapid integration of voice capabilities, and can be widely used in fields such as smart assistants and automated customer service.

Disadvantages：
The data source may have certain limitations. The VoiceAssistant-400K dataset is synthesized by GPT-4o and lacks real corpus verification, which may lead to insufficient generalization of the model to real voice environments. Future research may consider using richer actual voice datasets to further enhance the performance of the model in real voice tasks.

---

### Official Review · ~ChenJian1 · 2024-11-12
**Brief review**

**Rating:** 9
**Confidence:** 3

**Review:**

The paper introduces the Mini-Omni model, which is a significant innovation in the field of real-time voice interaction. The end-to-end design of the model reduces reliance on additional TTS systems, thereby reducing latency and improving user experience. The "Any Model Can Talk" approach provides an effective way for other models to quickly develop voice capabilities. Furthermore, the introduction of the VoiceAssistant-400K dataset provides a valuable resource for the fine-tuning of voice models. However, the complexity and resource requirements of the model are weaknesses that need to be further optimized in future work.